# Relevant Work Factors Associated with Voice Disorders in Early Childhood Teachers: A Comparison between Kindergarten and Elementary School Teachers in Yancheng, China

**DOI:** 10.3390/ijerph17093081

**Published:** 2020-04-28

**Authors:** Yaping Tao, Charles Tzu-Chi Lee, Yih-Jin Hu, Qiang Liu

**Affiliations:** 1Department of Preschool Education, Yancheng Teachers University, Yancheng 224002, China; 2Department of Health Promotion and Health Education, National Taiwan Normal University, Taipei 106, Taiwan

**Keywords:** vocal load, network questionnaire, Voice Handicap Index, daily class hours, types of voice usage

## Abstract

*Background:* Early childhood teachers consist of kindergarten and elementary school teachers in the lower grades. Young children at school may increase the vocal load of these teachers. Therefore, the objectives of this study were to determine the prevalence of voice disorders and the associated factors in early childhood teachers, and to determine if differences exist between kindergarten and elementary school teachers. *Method:* A cross-sectional survey was performed in July 2019 as a network questionnaire. Through cluster sampling, teachers (*n* = 414) from all five public kindergartens (*n* = 211) in the urban area of Yancheng, China, and four public elementary schools (*n* = 203) in the same school district participated in this study. Multivariate logistic regression models were used to analyze the associations among the prevalence of voice disorders in the teachers, school type, and relevant factors. *Results:* Our results indicated, based on the Voice Handicap Index scale (VHI-10, China), that the prevalence of voice disorders in early childhood teachers was 59.7%, while that in elementary school teachers (65.5%) was significantly higher than that in kindergarten teachers (54.0%) during the previous semester. Contributing factors included daily class hours, classroom air humidity, and speaking loudly during teaching. Additionally, certain types of voice usage in teaching such as falsetto speak, speaking more than other teachers, not using vocal techniques, and habitual voice clearing, were significantly associated with voice disorders. *Conclusion:* Most early childhood teachers have voice disorders. Compared with the kindergarten teachers, the elementary school teachers experienced a significantly higher prevalence of voice disorders. Several factors among work organization, work environment, and types of voice usage in teaching were associated with the voice disorders in early childhood teachers. The finding suggests that voice training should be provided for early childhood teachers, classroom teaching time should be decreased, and the number of teachers in basic subjects should be increased in the lower grades of elementary schools.

## 1. Introduction

In psychology, the term early childhood is usually defined as the time period from birth until the age of eight years, therefore covering infancy, kindergarten, and early school years up to Grade 3 (National Association for the Education of Young Children, NAEYC). At this stage, children show certain characteristics. They are full of curiosity and a desire to explore their world; however, their ability to care for themselves and demonstrate self-control is weak and they have not yet formed a sense of rules. Accordingly, in a school education setting, early childhood teachers often have to endure greater noise from the children, and, in turn, the teachers use more tactics related to oral management and guidance. A study that recorded the typical sounds present in kindergartens reported that the background noise in kindergartens is high and the voices of the teachers at work are significantly higher than the baseline [1]. In a similar study, the vocal load of kindergarten teachers was higher than that of elementary school teachers [2].

With regard to voice disorders in teachers, it has been reported that the younger the students, the higher the risk of developing a voice disorder [3]. Additionally, compared with middle school teachers, elementary school teachers and kindergarten teachers have been shown to have a higher risk of voice disorders [3,4,5,6,7]. In another study of elementary school teachers, the prevalence of voice disorders among teachers of the lower grades was greater than those of the higher grades [8]. Furthermore, teachers who taught Grade 4 or below had a higher risk of voice disorders than those who taught Grade 5 or above [9]. Compared with elementary school teachers, kindergarten teachers had a significantly higher risk of voice disorders [10]. However, some studies have not found an association between the grade taught and the prevalence of voice disorders in teachers [11,12,13,14]. One reason for a higher prevalence of voice disorders among lower grade teachers may be that early childhood teachers are primarily female, with females known to be more prone to voice disorders compared with males [15]. Therefore, the relationship between the student grade and teacher voice disorders cannot be confirmed without an adjustment for confounders, such as gender, age, and seniority.

According to student age, the six grades of elementary school are often divided into several levels, including the lower grades (1st and 2nd grade), middle grades (3rd and 4th grade), and higher grades (5th and 6th grade); or the lower grades (1st through 3rd grade) and higher grades (4th through 6th grades), depending on the educational strategy of the school. As such, the teachers of the different grades will have different vocal loads. Moreover, elementary school teachers often have their own subjects to teach. Studies have shown that physical education teachers [16,17] and music teachers [18] have a higher risk of voice disorders than teachers of other subjects. In China, it has also been shown that the prevalence of voice disorders in elementary school teachers of basic subjects (Chinese, mathematics, and English) is substantially higher than those of secondary subjects [19]. Teachers of basic subjects make up the majority of elementary school teachers, representing greater numbers than teachers of secondary subjects such as physical education and music.

Therefore, it is considered necessary to study special teacher groups to determine the prevalence of voice disorders and to clarify the risk factors. However, to date, the majority of studies focused on the development of voice disorders in teachers have been based on the school level without taking into account the age of the students. Although early childhood teachers come from different school levels, they should be considered as a special group worth studying in terms of voice disorder prevalence.

In mainland China, children under three years old are mostly educated by their families, with children at three to six years old entering kindergarten. Kindergarten education is a combination of caring and teaching, with the implementation of an integrated curriculum, including games as a basic activity [20]. Therefore, a kindergarten teacher instructs the children in all courses as a general teacher. At seven years old, children enter elementary schools, which implement departmental teaching, with the basic form of education being classroom teaching [21]. A teacher in elementary school usually teaches one course. Thus, the work environment and the education model used by kindergarten teachers and elementary school teachers in the lower grades are quite different, despite the fact that both types of teachers are considered early childhood educators.

In previous studies, together with individual factors, various work factors have been associated with the development of teacher voice disorders. These include work environment, such as noise, air quality, and sound equipment [13,22,23,24,25,26,27,28,29,30,31]; work organization [6,15,26,29,30], such as seniority, subject, grade, class size; and number of lessons [6,12,13,14,17,22,23,26,30,31,32,33,34,35,36,37] and types of voice usage in teaching [14,17,18,30,31,37]. Together, these factors have been reported to be associated with the development of voice disorders in teachers. Therefore, it is necessary to compare kindergarten teachers with elementary school teachers in the lower grades to determine who is more at risk and which work-related factors are risk factors for the voice disorders of early childhood teachers. This would be of great significance for intervention research in the future.

The objectives of our study were to investigate the prevalence of voice disorders in early childhood teachers, to compare the voice disorders and the relevant work factors between kindergarten teachers and elementary school teachers in the lower grades, and to identify the associations among work factors in early childhood teachers.

## 2. Materials and Methods

This study was a cross-sectional survey that was administered through a network questionnaire. This study obtained approval from the Research Ethics Committee of Taiwan Normal University, REC no. 201907HS011.

### 2.1. Sampling of Participants

The survey was conducted in July 2019, just as the teachers finished a semester of work and entered the summer vacation. We hypothesized that the time after a period of work, but before the rest period, would be the most helpful time for evaluating the voice conditions of the teachers. Teachers of all five public kindergartens in the urban area of Yancheng City, Jiangsu Province, China, and four public elementary schools in the same school district with kindergartens participated in the survey. The kindergarten teachers who participated in the study were full-time teachers; class childcare workers and kindergarten administrators, such as the kindergarten headmaster and the dean of studies, who are rarely directly involved in teaching activities, were excluded. Elementary school teachers who participated in the study were teachers in the lower grades of elementary school (1st through 4th grade), and they taught basic subjects such as Chinese, mathematics, and English. All teachers must have been considered to be on duty from February to June 2019, representing exactly one semester, without taking more than one week off. Finally, all qualified teachers of the nine schools agreed to participate in the survey and a total of 429 questionnaires were collected, of which 414 (96%) were validated. Among these, 211 were from kindergarten teachers and 203 were from elementary school teachers. 

### 2.2. Measurements

Based on previous studies and pre-interviews of the teachers participating this study, the study questionnaire was developed to include the following five sections:

(1) Background information, including gender, age, marital status (single/married, married means married, widowed or divorced), number of minor children, and level of education.

(2) Voice disorders, as four subsections: (a) Voice disease (Yes/No) and the type of voice disease in the previous five months (i.e., the previous semester), with 11 voice diseases provided to select (participants were required to report the voice diseases diagnosed by doctors when taking medical examinations or seeing a doctor at the hospital during the previous semester); (b) Voice disease (Yes/No) and the type of voice disease during the career, with 11 voice diseases provided to select; (c) Self-reported voice symptoms (Yes/No) and the type of voice symptoms in the previous semester, with 27 voice symptoms provided to select; and (d) Voice Handicap Index (VHI-10, China). The Voice Handicap Index is primarily used to evaluate the effect of voice disorders on individual quality of life. The scale used in this study, namely, the Voice Handicap Index with 10 items (VHI-10, China) was a simplified Chinese version of the Voice Handicap Index scale with 30 items (VHI-30) [38,39]. The reliability of the scale in terms of Cronbach’s α has been reported to be 0.94 [38]. The VHI-10 scale is a five-point scale as follows: 0 (never); 1 (rarely); 2 (sometimes); 3 (often); and 4 (always). The total score is 0–40, with the higher the score, the more serious the voice disorder. According to Arffa et al. [40], the cut-off point of the VHI-10 total score to define voice disorders is 11, thus several subsequent studies focused on voice disorders have adopted this criterion [28,32]. Therefore, in the current study, a teacher with a VHI-10 >11 was judged to have a voice disorder, while a teacher with a VHI-10 ≤11 was judged not to have a voice disorder.

(3) Work organization, including the number of students per class, daily class hours, years of teaching, being the head teacher or not, having other duties in school or not, daily teaching hours, daily hours outdoors with the students, and having voice training or not.

(4) Work environment. The teachers were asked to answer Yes or No with regard to 10 physical environments at school that may interfere with the teacher’s voice. These included: (a) Student noise, (b) Classroom echo, (c) Other noise in and out of the classroom and a school that may affect teaching, (d) Inconvenient amplification, (e) Bad audio quality of amplification, (f) Irritant smell in the classroom, (g) Smog, (h) Air humidity in the classroom, (i) A large space for the classroom (j), A large space for outdoor activity.

(5) Types of voice usage in teaching. Participants were asked to report Yes or No with regard to the use of six types of bad voice usage, as follows: (a) Falsetto speak; (b) Speaking loudly; (c) Speaking more than other teachers; (d) Not using vocal techniques; (e) Yelling when feeling emotional; and (f) Habitual voice clearing.

### 2.3. Statistical Analysis

The Pearson chi-square test and *t*-tests were applied to compare the prevalence of voice disorders, voice diseases, voice symptoms, and all work factors among the kindergarten and elementary school teachers. Multivariate logistic regression models were used to analyze the associations between the prevalence of voice disorders in the teachers, school type, and relevant factors. The strength of the associations was shown by odds ratios. Firstly, we performed regression analysis by adding background variables, work organization variables, work environment variables, and types of voice usage in teaching into the models. In other words, we analyzed different regression models in the following manner: (1) model 1: school type + background variables; (2) model 2: model 1 + work organization variables; (3) model 3: model 2 + work environment variables; (4) model 4: model 3 + types of voice usage variables. Secondly, multivariate logistic regression with forward selection was used to reveal the most significant explanatory variables for each of the models. In this study, SPSS 23.0 software (IBM Corp., Armonk, NY, USA) was used for data analysis. The significance level was set at 0.05. 

## 3. Results

### 3.1. Background Characteristics of the Participants

The majority of the 414 participants were female (94.9%). The mean (SD) age was 33.4 (7.5), with the age of the participants primarily distributed in the 30–39 (40.6%) and 20–29 (36.5%) age groups. The majority (80.7%) of the participants were or had been married, including both divorced and widowed participants. About half (51.4%) had a minor child, 32.1% had no minor children, and 16.4% had more than one child. Most (65.7%) participants had a college degree, while the others had a bachelor’s degree or above. There were no significant differences in terms of age, marital status, or number of minor children between the elementary school teachers and kindergarten teachers; however, the proportion of kindergarten teachers that were female was significantly higher than that for elementary school teachers (χ^2^ = 9.019, *p* = 0.003). The education level of the elementary school teachers was significantly higher than that of the kindergarten teachers (χ^2^ = 50.673, *p* < 0.001) (Table 1).

### 3.2. Voice Conditions

Most (59.7%) of the participants had scores >11, indicating they had a voice disorder. Additionally, 70% of the participants reported having been diagnosed with a voice disease during the past semester. Furthermore, 78.3% of the participants reported having been diagnosed with a voice disease during their career. Only 7.2% of the participants had no voice symptoms. Compared with the kindergarten teachers, the elementary school teachers experienced a significantly higher prevalence of voice disorders (χ^2^ = 5.674, *p* = 0.017), voice diseases (χ^2^ = 8.776, *p* = 0.003), and voice symptoms (χ^2^ = 6.475, *p* = 0.011) over the previous semester. However, no statistical differences were observed for the prevalence of voice diseases during the course of their career (Table 2).

The average prevalence of the 11 voice diseases in early childhood teachers from high to low were chronic pharyngitis (40.3%), chronic laryngitis (23.2%), acute pharyngitis (16.7%), vocal nodules (15%), vocal fold congestion (11.8%), acute laryngitis (8.7%), vocal fold edema (5.8%), vocal fold polyps (5.3%), vocal hypertrophy (3.9%), vocal fold bleeding (3.6%), and incomplete vocal fold closure (1.2%). Figure 1 shows the distribution of voice diseases and voice symptoms among the kindergarten teachers and elementary school teachers. The prevalence of reported voice diseases in the teachers was higher during the course of their career than for the past semester. The top 15 of the 24 voice symptoms in all teachers were hoarseness (65.7%), dryness (59.9), tired voice (57.5%), ache (47.6%), scratchiness (41.3%), foreign body sensation (29.5%), frequent throat clearing (27.3%), pronunciation difficulty (26.8%), strain (24.6%), coughing easily when speaking (23.4%), phlegm (21.5%), high note difficulty (17.1%), don’t like to talk (15.7%), swallowing difficulty (14.3%), and loss of voice (14%). Compared with kindergarten teachers, elementary school teachers had a higher prevalence of 8 diseases and 14 of the top 15 voice symptoms. 

### 3.3. Work Factors

Among the work organization factors, there was only one factor, namely, being head teacher or not (χ^2^ = 0.129, *p* > 0.72), that showed no significant difference between the kindergarten teachers and elementary school teachers. The number of students per class (t = −50.115, *p* < 0.001), daily class hours (t = −29.060, *p* < 0.001), years of teaching (χ^2^ = 8.989, *p* = 0.003), having other duties in school (χ^2^ = 7.818, *p* = 0.005), daily teaching hours (χ^2^ = 12.587, *p* = 0.002), and having voice training (χ^2^ = 8.319, *p* = 0.004) were significantly greater among elementary school teachers compared with kindergarten teachers. For the kindergarten teachers, the daily hours spent outdoors with students (χ^2^ = 229.177, *p* < 0.001) were significantly greater than those for elementary school teachers (Table 3).

Among the work environment factors, elementary school teachers reported significantly higher ratios than kindergarten teachers in terms of inconvenient amplification (χ^2^ = 4.030, *p* = 0.045), irritant smell in the classroom (χ^2^ = 7.310, *p* = 0.007), smog (χ^2^ = 12.990, *p* < 0.001), and air humidity in the classroom (χ^2^ = 26.155, *p* < 0.001). No other work environment factors showed a significant difference between the two groups of teachers. The work environments that more than half of the participants selected to have a negative impact on their voices included student noise (80.7%), other noise (62.3%), inconvenient amplification (84.5%), smog (64.3%), air humidity in the classroom (50.2%), and a large space for outdoor activity (69.3%) (Table 3).

Among the types of voice usage in teaching, only speaking loudly showed a significant difference between the elementary school teachers and kindergarten teachers (χ^2^ = 7.310, *p* = 0.001). More than half of the participants selected yes to raising their voice (82.6%), speaking more than other teachers (79.7%), not using vocal techniques (85.7%), yelling when feeling emotional (73.2%), and habitual voice clearing (60.9%) (Table 3).

### 3.4. Associated Factors with Voice Disorders

In Table 4, univariate analysis of logistic regression revealed the independent variables associated with voice disorders. School type, daily class hours, daily teaching hours, voice training, student noise, classroom echo, other noise, irritant smell in the classroom, smog, air humidity in the classroom, a large space for the classroom, a large space for outdoor activity, falsetto speak, speaking loudly, speaking more than other teachers, not using vocal techniques, yelling when feeling emotional, and habitual voice clearing were all associated with voice disorders in the teachers. 

Adjusted model 1 presented the multivariate analysis of the logistic regression with the variables school type and five background factors. Adjusted model 2 added eight work organization factors to the variables of model 1. Adjusted model 3 added 10 work environment factors to the variables of model 2. Adjusted model 4 added six types of voice usage in teaching to the variables of model 3.

Compared with the unadjusted value (OR = 1.62; 95% CI, 1.09–2.04; *p* = 0.018), the strength of the association between the school type and voice disorders decreased gradually and the significance disappeared after model 1 (OR = 1.45; 95% CI, 0.93–2.25; *p* = 0.099) to model 4 (OR = 0.87; 95% CI, 4.15–4.94; *p* = 0.876) adjustments, which were explained by factors from work organization (daily class hours, daily teaching hours, daily hours outdoors with students), work environments (other noise, air humidity in classroom, large space for outdoor activity), and types of voice usage (falsetto speak, speaking loudly, speaking more than other teachers, not using vocal techniques, habitual voice clearing). 

Model 4, the full adjusted model, showed daily class hours (OR = 1.68; 95% CI, 1.10–2.56; *p* = 0.016), air humidity in the classroom (OR = 2.94; 95% CI, 1.50–5.76; *p* = 0.002), falsetto speak (OR = 2.16; 95% CI, 1.13–4.13; *p* = 0.02), speaking loudly (OR = 2.47; 95% CI, 1.20–5.08; *p* = 0.014), speaking more than other teachers (OR = 2.13; 95% CI, 1.10–4.11; *p* = 0.025), not using vocal techniques (OR = 2.85; 95% CI, 1.37–5.96; *p* = 0.005), and habitual voice clearing (OR = 3.06; 95% CI, 1.78–5.25; *p* < 0.001) were associated with voice disorders in early school teachers.

Compared with the regression analysis above, multivariate logistic regression with forward selection presented similarly significant variables associated with the development of voice disorders in the teachers. In model 2: daily class hours (OR = 1.51; 95% CI, 1.09–2.11; *p* = 0.015) and daily teaching hours ≥ 7 (OR = 1.91; 95% CI, 1.15–3.17; *p* = 0.013) were significantly associated. In model 3: other noise (OR = 2.21; 95% CI, 1.40–3.49; *p* = 0.001), air humidity in the classroom (OR = 2.37; 95% CI, 1.47–3.81; *p* < 0.001), and a large space for outdoor activity (OR = 1.97; 95% CI, 1.21–3.20; *p* = 0.006) were significantly associated. In model 4: daily class hours (OR = 1.51; 95% CI, 1.02–2.22; *p* = 0.038), other noise (OR = 1.72; 95% CI, 1.05–2.82; *p* = 0.031), air humidity in the classroom (OR = 2.84; 95% CI, 1.72–4.67; *p* < 0.001), falsetto speak (OR = 2.27; 95% CI, 1.24–4.13; *p* = 0.008), speaking loudly (OR = 2.33; 95% CI, 1.22–4.43; *p* = 0.01), speaking more than other teachers (OR = 2.15; 95% CI, 1.19–3.89; *p* = 0.011), not using vocal techniques (OR = 2.60; 95% CI, 1.32–5.10; *p* = 0.006), and habitual voice clearing (OR = 2.47; 95% CI, 1.53–3.99; *p* < 0.001) were significantly associated. 

## 4. Discussion

Based on the VHI-10 scale (China), the prevalence of voice disorders in early childhood teachers was 59.7%, with an average VHI-10 score in this study of 13.5. In comparison, 28.8% of Brazil’s primary school and kindergarten teachers [28] and 10.4% of Malaysia’s middle school teachers [32] have been reported to have a voice disorder. In contrast, an average VHI-10 score of 10 has been reported for kindergarten teachers in Greece [41]. Therefore, in this study, we found a higher prevalence of voice disorders in early childhood teachers in China. However, the prevalence of voice disorders was even higher in Egypt’s middle school and university teachers, being 80.1% [35].

Regarding the self-reported diagnosed voice diseases of early childhood teachers, the prevalence in the previous semester was as high as 70%, with kindergarten teachers at 63.5% and elementary school teachers at 76.8%. These results are similar to the survey results (65%) of kindergarten teachers in Wenzhou, China [42], but the incidence of voice diseases during the course of the career (78.3%) is far higher than that of Spanish teachers (16.4%) [17]. In general, the prevalence of voice diseases in teachers in most studies has been obtained by clinical examination at a certain point in time, and has ranged from 10.9% to 34% [11,19,43,44]. Our conclusion is consistent with a previous review of voice disorders in teachers, in which a longer recall period would be expected to result in a higher prevalence of voice disorders than a shorter recall period [15]. Chronic pharyngitis and chronic laryngitis represented the two diseases with the highest prevalence in this study, which is consistent with clinically verified findings [19,45,46]. It is worth noting that the prevalence of vocal fold nodules in teachers over one semester reached up to 11.8%, which is similar to the findings of Lv Dan et al. of 389 elementary school teachers [44], but much higher than many large-scale surveys that ranged from 1.55% to 2.58% [11,19,46,47]. Vocal nodules, chronic laryngitis, and vocal fold congestion were three diseases with a high prevalence in this study. These diseases are associated with the typical type of vocal fold lesions that are the result of bad voice behaviors, including the voice being abused for a long time and a lack of correct voice skills, which are common underlying causes of voice disorders [48]. Previous studies have shown that voice training is the basic treatment for these diseases [49,50].

This is the first study that allowed for 24 kinds of voice symptoms to be selected by the teachers, a decision based on teacher interviews and previous studies. In this manner, we believed that providing more symptoms to select would result in a more accurate determination of the voice disorder. Our results showed that every voice symptom was selected by the early childhood teachers, with 92.8% of the teachers having one or more voice symptoms, which is higher than the highest proportion reported (80%) from similar studies [51]. Among the 24 voice symptoms, hoarseness had the highest selection rate, which is consistent with other surveys. Similarly, the symptoms that ranked in the top 15 selected by the early childhood teachers were also the common symptoms found by several previous studies [14,22,34,35,52]. Some teachers stated in the interview that professional teachers always have voice disorders and it is very common to have one or more voice symptoms. Voice symptoms are not only the most direct manifestation of voice disorders, but they also are precursors of voice disorders. Therefore, teachers should pay attention to vocal symptoms, particularly when they become greater or last longer; thus, timely diagnosis and treatment are important.

To summarize, most teachers have voice disorders, voice diseases, and voice symptoms, and the rate is close to or higher than that in the previous research, which shows that the voice health of early childhood teachers in Yancheng, China is poor. In addition, teachers are familiar with this problem, and it may have a negative impact on every teacher and the entire education industry. Thus, education departments and occupational safety and health departments should attach importance to the voice disorders of early childhood teachers and put forward effective strategies to improve their voice health.

One purpose of this study was to determine if there was a difference in voice disorders between the kindergarten and elementary school teachers. After statistical analysis, we found that during the course of their career, there was no difference in the voice disorders between the two types of teachers; however, in the previous semester, the prevalence of voice disorders and voice symptoms in the elementary teachers was significantly higher than that for kindergarten teachers. These results are not consistent with the prediction that the younger the students, the higher the risk of voice disorders in the teacher. Additional multivariate logistic regression analysis showed that the difference in the prevalence of voice disorders between the kindergarten and elementary school teachers was primarily due to various work factors, including: 

(1) Daily class hours: Table 2 shows that the daily class hours for the kindergarten teachers were much less than for the elementary school teachers (0.77 vs. 2.57). Additionally, Table 4 shows that when the class time was increased by one hour, the teacher’s risk of a voice disorder increased 0.68 times. Other studies, including that by Pizolato et al. [6], Fu et al. [11], and Lee [33], have also reported that weekly class hours were significantly correlated with the development of voice disorders in the teachers. 

(2) Air humidity in the classroom: A greater number of elementary school teachers (63.1%) reported air humidity affecting vocal use in teaching compared with kindergarten teachers (37.9%). Among these teachers, those who reported air humidity had a higher risk of a voice disorder (OR = 2.94) compared with those who did not report this parameter. This conclusion is consistent with studies by Cutiva et al. [25] and Sampaio et al. [28] using unadjusted regression analysis. However, after adjusted regression, air humidity was not associated with voice disorders in the teachers in either study. 

(3) Types of voice usage in teaching: Among the five types of voice usage associated with voice disorders, speaking loudly was significantly more common in elementary school teachers than in kindergarten teachers (89.2% vs. 76.3%). In addition, we found that the risk of voice disorders was 2.47 times higher in teachers who spoke loudly during teaching. Similar results have been reported by several studies, including those by Fu et al. [11], Chen et al. [14], Ubillos et al. [17], Trinite et al. [23], and Lin et al. [53].

As early as 2014, the Jiangsu province of China started the reform of the kindergarten play-orientated curriculum [54]. Subsequently, from 2018, the ministry of education of China began to correct errors regarding the tendency of pre-school education to mimic that of elementary school, thus preventing kindergartens from adopting departmental teaching and taking classroom teaching as the main form similar to elementary school [55]. Under the influence of these policies, the number of classroom lessons in a given kindergarten was reduced from three or four per day to one or two per day, with the main activities for children being games in small groups. Accordingly, the main tasks of the kindergarten teachers are to provide supportive educational resources, observe the children, and provide individualized help and guidance when needed. It should be noted that all the kindergartens in this study were standardized, with two teachers, one nurse, and about 30 children per class. After these new policies, the teachers in these kindergarten classes spent less time on oral teaching and did not have to speak loudly all the time. According to several kindergarten teachers who were interviewed as part of this study, the teachers felt that the vocal load was much reduced compared with the previous educational policies. However, elementary school teachers in the lower grades, who also teach young children, are being tasked with as many as four lessons (40 min/lesson) per day requiring oral teaching. As such, these teachers typically have to speak loudly to be heard by all the pupils. In summary, elementary schools were found to have substantially more work factors associated with voice disorders than the kindergartens in our study. 

Teachers in this study were in the same geographic area, thus the air humidity in the classroom may not have been different between the kindergartens and elementary schools. However, a greater number of elementary school teachers felt that the air humidity affected their voice, with nearly 60% of the teachers reporting dryness of the throat. We consider that this may be connected with the heavy vocal load of the elementary school teachers. As such, when voice fatigue and damage occur, the teachers may be more sensitive to air humidity.

Falsetto speak, speaking loudly, speaking more than other teachers, not using vocal techniques, and habitual voice clearing were all associated with the prevalence of voice disorders in the early childhood teachers in our study. Several studies have reported similar conclusions [11,14,17,22,23,42,53]. We suspect that one reason why so many teachers use their voice in unhealthy ways may be associated with the lack of voice training, because some studies have confirmed the correlation between voice training and a reduction in voice disorders [7,56]. The majority (91.5%) of the teachers in our study reported never having received any kind of voice training. However, numerous studies of voice disorders have suggested voice training for professional voice users in order to maintain or improve their voices. 

## 5. Limitations

There are several limitations of this study. The study used a cross-sectional design, thus, no causal relationships should be assumed. With the exception of the VHI scale, which is standardized, most of the data were from the subjective reports of the participants, which may have led to some research errors. The 414 participants in this study were taken from public schools in the urban area of Yancheng, China, thus representing the voice conditions of only some teachers in one region of China. Therefore, the results cannot be extended to other regions or to private school teachers.

## 6. Conclusions

Most early childhood teachers have voice disorders. The prevalence of voice disorders in elementary school teachers was significantly higher than that in kindergarten teachers. This higher prevalence was determined to be associated with various work factors, including daily class hours, air humidity in the classroom, and speaking loudly during teaching. Recent reform in the Chinese curriculum used in kindergartens has reduced the vocal load of kindergarten teachers to a certain extent. However, teachers in the lower grades of elementary schools, who are also early childhood teachers, have a high vocal load because of longer lesson hours requiring more oral teaching. Our results also indicated that a number of particular types of voice usage in teaching were associated with voice disorders in early childhood teachers, but the majority of the teachers have not received any proper voice training. A number of studies have confirmed that voice disorders can be prevented [57,58]. Additionally, voice training has been shown to effectively improve vocal hygiene knowledge and consciousness and quality of voice, and alleviate uncomfortable voice symptoms [59]. We also consider that education departments and occupational safety and health departments should focus more on the voice health of teachers and provide voice training regarding vocal hygiene and vocal techniques. Additionally, in order to reduce the vocal load of elementary school teachers in the lower grades, classroom teaching time should be decreased and the number of teachers in basic subjects should be increased.

## Figures and Tables

**Figure 1 ijerph-17-03081-f001:**
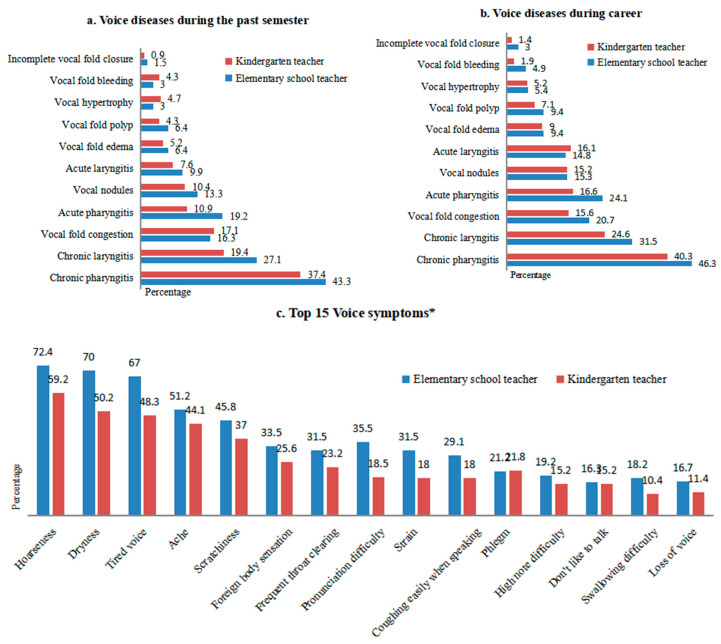
(**a**) Voice diseases during the past semester, (**b**) Voice diseases during career, (**c**) Top 15 Voice symptoms. Comparison of voice diseases and voice symptoms between kindergarten and elementary school teachers, Yancheng City, Jiangsu Province, China (2019), *n* = 414. Note: * Teachers were asked to select from 24 voice symptoms; the other 9 were pronunciation weakness, headache when speaking, tone anomaly, sore neck and shoulder when speaking, volume anomaly, whispering, breathing fast when speaking, low note difficulty, and loss of sensation in the throat.

**Table 1 ijerph-17-03081-t001:** Distribution of the background characteristics of participants.

Characteristics	Number (%)			χ^2^	*p*-Value
	Total *n* = 414	Kindergarten *n* = 211	Elementary School *n* = 203		
**Gender**					
Female	393 (94.9)	207 (98.1)	186 (91.6)	9.019	0.003 **
Male	21 (5.1)	4 (1.9)	17 (8.4)		
**Age**					
20–29	151 (36.5)	85 (40.3)	66 (32.5)	2.724	0.436
30–39	168 (40.6)	80 (37.9)	88 (43.3)		
40–49	85 (20.5)	41 (19.4)	44 (21.7)		
≥50	10 (2.4)	5 (2.4)	5 (2.5)		
**Marital status**					
Single	80 (19.3)	36 (17.1)	44 (21.7)	1.412	0.235
Married	159 (80.7)	175 (82.9)	159 (78.3)		
**Minor children**					
0	133 (32.1)	66 (31.3)	67 (33.0)	0.233	0.890
1	213 (51.4)	(52.6)	102(50.2)		
>1	68 (16.4)	34 (16.1)	34 (16.7)		
**Education**					
College degree	272 (65.7)	173 (82.0)	99 (48.8)	50.673	0.000 ***
Bachelor degree and above	142 (34.3)	38 (18.0)	104 (51.2)		

Note: Married = married /widowed/divorced; ** *p* < 0.01; *** *p* < 0.001.

**Table 2 ijerph-17-03081-t002:** Prevalence of voice disorders in early childhood teachers, Yancheng City, Jiangsu Province, China (2019), *n* = 414.

Voice Disorders	Number (%)			χ^2^	*p*-Value
	Total *n* = 414	Kindergarten *n* = 211	Elementary School *n* = 203		
**Voice disorders**					
Yes (VHI-10 score >11)	247 (59.7)	114 (54.0)	133 (65.5)	5.674	0.017 *
No (VHI-10 score ≤11)	167 (40.3)	97 (46.0)	70 (34.5)		
**Voice diseases**					
During last semester					
Yes	290 (70.0)	134 (63.5)	156 (76.8)	8.776	0.003 **
No	124 (30.0)	77 (36.5)	47 (23.2)		
During Career					
Yes	324 (78.3)	158 (74.9)	166 (81.8)	2.888	0.089
No	90 (21.7)	53 (25.1)	37 (18.2)		
**Voice symptoms**					
Yes	384 (92.8)	189 (89.6)	195 (96.1)	6.475	0.011 *
No	30 (7.2)	22 (10.4)	8 (3.9)		

Note: * *p* < 0.05, ** *p* < 0.01.

**Table 3 ijerph-17-03081-t003:** Distribution analysis of work-related factors associated with voice disorders in early childhood teachers, Yancheng City, Jiangsu Province, China (2019), *n* = 414.

Variables	Number (%)/Mean ± SD			χ^2^/t	*p-*Value
	Total *n* = 414	Kindergarten *n* = 211	Elementary School *n* = 203		
**Work organization**					
Students per class	41.4 ± 12.0	30.5 ± 3.7	52.7 ± 5.1	−50.115 ^a^	0.000 ***
Daily class hours	1.7 ± 1.1	0.77 ± 0.4	2.6 ± 0.8	−29.060 ^a^	0.000 ***
Years of teaching					
≤10	188 (45.4)	111 (52.6)	77 (37.9)	8.989	0.003 **
>10	226 (54.6)	100 (47.4)	126 (62.1)		
Head teacher					
No	226 (54.6)	117 (55.5)	109 (53.7)	0.129	0.720
Yes	188 (45.4)	94 (44.5)	94 (46.3)		
Other duties in school					
0	333 (80.4)	181 (85.8)	152 (74.9)	7.818	0.005 **
≥1	81 (19.6)	30 (14.2)	51 (25.1)		
Daily teaching hours					
≤4	104 (25.1)	53 (25.1)	51 (25.1)	12.587	0.002 **
4–6	146 (35.3)	90 (42.7)	56 (27.6)		
≥7	164 (39.6)	68 (32.2)	96 (47.3)		
Daily hours outdoors with students					
0–1	206 (49.8)	28 (13.3)	178 (87.7)	229.177	0.000 ***
1–2	143 (34.5)	126 (59.7)	17 (8.4)		
≥3	65 (15.7)	57 (27)	8 (3.9)		
Voice training					
No	379 (91.5)	185 (87.5)	194 (95.6)	8.319	0.004 **
Yes	35 (8.5)	26 (12.3)	9 (4.4)		
**Work environments (Yes)**					
Student noise	334 (80.7)	170 (80.6)	164 (80.8)	0.003	0.955
Classroom echo	84 (20.3)	43 (20.4)	41 (20.2)	0.002	0.963
Other noise	258 (62.3)	122 (57.8)	136 (67.0)	3.709	0.054
Inconvenient amplification	350 (84.5)	171 (81.0)	179 (88.2)	4.030	0.045 *
Bad audio quality of amplification	203 (49.0)	97 (46.0)	106 (52.2)	1.615	0.204
Irritant smell in classroom	78 (18.8)	29 (13.7)	49 (24.1)	7.310	0.007 **
Smog	266 (64.3)	118 (55.9)	148 (72.9)	12.990	0.000 ***
Air humidity in classroom	208 (50.2)	80 (37.9)	128 (63.1)	26.155	0.000 ***
Large space for classroom	205 (49.5)	97 (46.0)	108 (53.2)	2.164	0.141
Large space for outdoor activity	287 (69.3)	148 (70.1)	139 (68.5)	0.136	0.713
**Types of voice usage in teaching (Yes)**					
Falsetto speak	101 (24.4)	55 (26.1)	46 (22.7)	0.651	0.420
Speak loudly	342 (82.6)	161 (76.3)	181 (89.2)	11.908	0.001 **
Speak more than other teachers	330 (79.7)	167 (79.1)	163 (80.3)	0.084	0.771
Not use vocal techniques	355 (85.7)	186 (88.2)	169 (83.3)	2.033	0.154
Yell when feeling emotional	303 (73.2)	151 (71.6)	152 (74.9)	0.579	0.447
Habitually Voice clearing	252 (60.9)	128 (60.7)	124 (61.1)	0.008	0.930

Note: ^a^
*t*-test: * *p* < 0.05, ** *p* < 0.01, *** *p* < 0.001.

**Table 4 ijerph-17-03081-t004:** Multivariate logistic regression analysis for voice disorders in early childhood teachers, Yancheng City, Jiangsu Province, China (2019), *n* = 414.

Variable	Unadjusted Model ^a^		Adjusted Model 1 ^b^		Adjusted Model 2 ^c^		Adjusted Model 3 ^d^		Adjusted Model 4 ^e^	
	OR (95% CI)	*p-*Value	OR (95% CI)	*p-*Value	OR (95% CI)	*p-*Value	OR (95% CI)	*p-*Value	OR (95% CI)	*p-*Value
**School type**										
Kindergarten	Reference		Reference		Reference		Reference		Reference	
Elementary school	1.62 (1.09–2.40)	0.018 *	1.45 (0.93–2.25)	0.099	1.38 (0.34–5.65)	0.657	1.21 (0.26–5.62)	0.807	0.87 (0.15–4.94)	0.876
**Background**										
Gender										
Female	Reference		Reference		Reference		Reference		Reference	
Male	0.90 (0.37–2.18)	0.809	0.71 (0.29–1.79)	0.470	0.75 (0.29–1.94)	0.556	0.66 (0.24–1.8)	0.411	0.79 (0.26–2.37)	0.668
Age										
20–29	Reference		Reference		Reference		Reference		Reference	
30–39	1.03 (0.66–1.61)	0.906	1.14 (0.62–2.07)	0.681	1.35 (0.63–2.93)	0.443	1.72 (0.73–4.02)	0.214	1.71 (0.69–4.22)	0.246
40–49	1.31 (0.76–2.28)	0.333	1.64 (0.84–3.23)	0.150	2.07 (0.80–5.37)	0.133	2.09 (0.74–5.91)	0.165	2.97 (0.97–9.13)	0.058
≥50	0.72 (0.20–2.58)	0.609	1.00 (0.25–3.97)	0.996	0.93 (0.20–4.44)	0.932	1.38 (0.25–7.58)	0.712	1.19 (0.21–6.86)	0.842
Marital status										
Single	Reference		Reference		Reference		Reference		Reference	
Married	0.86 (0.52–1.43)	0.565	0.71 (0.33–1.51)	0.375	0.60 (0.27–1.32)	0.204	0.53 (0.23–1.25)	0.146	0.45 (0.18–1.14)	0.092
Minor children										
0	Reference		Reference		Reference		Reference		Reference	
1	1.01 (0.65–1.57)	0.967	1.18 (0.63–2.20)	0.607	1.17 (0.61–2.24)	0.636	1.09 (0.53–2.21)	0.819	1.13 (0.53–2.43)	0.756
>1	1.04 (0.57–1.89)	0.903	1.19 (0.54–2.63)	0.662	1.21 (0.53–2.78)	0.652	1.07 (0.43–2.66)	0.886	0.94 (0.35–2.47)	0.893
Education										
College degree	Reference		Reference		Reference		Reference		Reference	
Bachelor degree and above	1.45 (0.96–2.22)	0.081	1.37 (0.84–2.25)	0.205	1.36 (0.81–2.28)	0.247	1.08 (0.61–1.91)	0.798	1.00 (0.53–1.87)	0.987
**Work organization**										
Students per class	1.02 (1.00–1.03)	0.085			0.96 (0.91–1.01)	0.080	0.96 (0.91–1.01)	0.103	0.96 (0.91–1.02)	0.216
Daily class hours	1.38 (1.14–1.66)	0.001 **			1.64 (1.16–2.32)	0.006 **	1.49 (1.03–2.17)	0.035 *	1.68 (1.10–2.56)	0.016 *
Years of teaching										
≤10	Reference				Reference		Reference		Reference	
>10	1.05 (0.71–1.56)	0.815			0.89 (0.43–1.84)	0.757	0.90 (0.41–1.98)	0.784	0.92 (0.39–2.13)	0.837
Head teacher										
No	Reference				Reference		Reference		Reference	
Yes	1.22 (0.82–1.81)	0.331			1.18 (0.77–1.8)	0.445	1.06 (0.67–1.68)	0.816	0.97 (0.59–1.61)	0.917
Other duties in school										
0	Reference				Reference		Reference		Reference	
≥1	1.11 (0.68–1.83)	0.673			1.12 (0.65–1.96)	0.679	0.89 (0.48–1.63)	0.705	0.98 (0.51–1.91)	0.957
Daily teaching hours										
≤4	Reference				Reference		Reference		Reference	
4–6	1.58 (0.95–2.62)	0.079			1.71 (1.00–2.92)	0.049 *	1.5 (0.84–2.68)	0.176	1.42 (0.75–2.67)	0.283
≥7	2.00 (1.21–3.31)	0.007 **			1.98 (1.16–3.35)	0.012 *	1.73 (0.97–3.09)	0.063	1.51 (0.81–2.82)	0.200
Daily hours outdoors with students										
0–1	Reference				Reference		Reference		Reference	
1–2	0.78 (0.50–1.20)	0.255			1.03 (0.54–1.99)	0.920	0.83 (0.41–1.68)	0.596	0.86 (0.41–1.84)	0.706
≥3	0.54 (0.31–0.96)	0.034 *			0.6 (0.28–1.28)	0.185	0.37 (0.16–0.85)	0.020 *	0.41 (0.17–1.00)	0.050
Voice training										
No	Reference				Reference		Reference		Reference	
Yes	0.65 (0.35–1.39)	0.301			0.79(0.37–1.65)	0.525	1.16 (0.49–2.73)	0.737	0.98(0.40–2.40)	0.965
**Work environments**										
Student noise (Y/N)	2.54 (1.54–4.18)	0.000 ***					1.27 (0.68–2.39)	0.450	1.12 (0.56–2.24)	0.740
Classroom echo (Y/N)	3.02 (1.72–5.31)	0.000 ***					1.52 (0.75–3.09)	0.244	1.19 (0.55–2.58)	0.651
Other noise (Y/N)	3.51 (2.31–5.32)	0.000 ***					1.86 (1.08–3.19)	0.025 *	1.46 (0.81–2.63)	0.214
Inconvenient amplification (Y/N)	1.48 (0.87–2.53)	0.152					1.40 (0.72–2.73)	0.327	1.60 (0.77–3.32)	0.207
Bad audio quality (Y/N)	1.32 (0.89–1.96)	0.168					0.97 (0.60–1.58)	0.913	0.88 (0.52–1.48)	0.623
Irritant smell in classroom (Y/N)	2.89 (1.62–5.16)	0.000 ***					1.14 (0.54–2.43)	0.728	1.43 (0.62–3.30)	0.403
Smog (Y/N)	2.53 (1.67–3.83)	0.000 ***					0.93 (0.52–1.67)	0.807	0.82 (0.44–1.55)	0.547
Air humidity in classroom (Y/N)	0.25 (0.17–0.38)	0.000 ***					2.45 (1.31–4.57)	0.005 **	2.94 (1.50–5.76)	0.002 **
Large space for classroom (Y/N)	2.76 (1.84–4.15)	0.000 ***					0.94 (0.53–1.67)	0.828	0.87 (0.46–1.65)	0.678
Large space for outdoor activity (Y/N)	3.21 (2.08–4.95)	0.000 ***					1.93 (1.09–3.43)	0.025 *	1.76 (0.93–3.34)	0.084
**Types of voice usage**										
Falsetto speak (Y/N)	2.89 (1.73–4.84)	0.000 ***							2.16 (1.13–4.13)	0.020 *
Speak loudly (Y/N)	4.04 (2.35–6.96)	0.000 ***							2.47 (1.20–5.08)	0.014 *
Speak more (Y/N)	3.69 (2.23–6.11)	0.000 ***							2.13 (1.10–4.11)	0.025 *
Not use vocal techniques (Y/N)	1.93 (1.11–3.37)	0.020 *							2.85 (1.37–5.96)	0.005 *
Yell when feeling emotional (Y/N)	2.51 (1.61–3.92)	0.000 ***							0.71 (0.38–1.34)	0.296
Habitually Voice clearing (Y/N)	3.39 (2.24–5.13)	0.000 ***							3.06 (1.78–5.25)	0.000 ***

Note: Abbreviation: OR, odds ratio; CI, confidence interval; * *p* < 0.05, ** *p* < 0.01, *** *p* < 0.001; ^a^ Unadjusted Model: Univariate logistic regression analysis of all independent variables; ^b^ Adjusted Model 1: Model of multivariate logistic regression with background variables; ^c^ Adjusted Model 2: Add work organization variables to model 1; ^d^ Adjusted Model 3: Add work environment variables to model 2; ^e^ Adjusted Model 4: Add types of voice usage in teaching to model 3.

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
