# Peer review of "Relevant Work Factors Associated with Voice Disorders in Early Childhood Teachers: A Comparison between Kindergarten and Elementary School Teachers in Yancheng, China"

_ijerph, 2020, doi:10.3390/ijerph17093081_

Round 1
Reviewer 1 Report
In general terms I found this study interesting although not very novel. There are several publications on the work-relatedness of voice disorders among teachers, which have determined that nowadays the tendency is moving to intervention studies (because there is an overdiagnosis of voice disorders among teachers). However, this study explored 24 symptoms among around 400 participants, which would add some knowledge to the topic, if the authors would explore more those results.
As of now, I found this study very similar to previous studies that explored associated factors of voice disorders among teachers (Factors associated with voice-related quality of life among teachers with voice complaints. LCC Cutiva, A Burdorf, Journal of communication disorders 52, 134-142; Work-related determinants of voice complaints among school workers: An eleven-month follow-up study. LCC Cutiva, A Burdorf, American journal of speech-language pathology 25 (4), 590-597). However, if the authors manage to explore the relationship of the work-related factors with the symptoms, and for instance determine which factors are strongly associated with the most common symptoms. That would be information that would allow to make epidemiologic decisions on this topic.
Please see more detailed comments below:
- Please check the reference style. There are some parts where references are between squared brackets but also in some parts are the numbers without brackets
- Page 2, line 25. Do you mean vocal load instead of voice load?
- “However, in order to determine who is more at risk and which work factors are considered greater risk factors for the development of voice disorders in early childhood teachers, it is necessary to compare kindergarten teachers with elementary school teachers in the lower grades.” Why?
The relationship between teaching and voice disorders have been widely reported. Although, comparisons per level of teaching have been performed mainly between primary and secondary teachers. It has been reported a higher likelihood among primary school teachers due to conditions as those mentioned by the authors in the introduction. However, I would hypothesize that kindergarten teachers would have less voice disorders because they groups may be smaller, and they may have lower “vocal demand”.
The Introduction is well written, but it is not clear why this study is needed
- Statistical analysis, since the information is self-reported what do the authors mean by “voice diseases”?
- Page 5, voice conditions. In the methods section the authors stated that “The Voice Handicap Index is primarily used to evaluate the effect of voice disorders on individual quality of life”; however, in page 5 they report that “The mean (SD) VHI-10 score was 13.5 (7.8), with most (59.7%) of the participants having scores > 11, indicating they had a voice disorder”. This is technically correct because speakers cannot feel an effect of voice disorders measured by means of the VHI-10, if they don’t have the voice disorders first. Nevertheless, as it is presented now may be confusing.
- Page 6, line 8, do the authors mean “organic voice disorders” when they mention “voice diseases”?
- Page 6, “The prevalence of reported voice diseases in the teachers was higher during the course of their career than for the past semester”. This is expected because the recall period is longer
- Page 7, line 10-13. The sentence is grammatically correct but may be difficult to follow. I suggest… The number of students per class (t = −50.115,p < 0.001), daily class hours (t = −29.060,p < 0.001), years of teaching (χ2 = 8.989,p = 0.003), having other duties in school (χ2 = 7.818,p = 0.005), daily teaching hours (χ2 = 12.587, p = 0.002), and having voice training (χ2 = 8.319,p = 0.004) were significantly greater among elementary school teachers compared with kindergarten teachers.
- Page 8, line 7. May be is better “groups of teachers” instead of “kinds of teachers”
- Page 8, line 7-10. Please check grammar of the sentence
- Page 8, line 12-13. More than 50% instead of “greater than half”
- Page 11, line 9-41. The first paragraph of this section mentions how the relationship between school type and “organic” voice disorders changed along the models. Therefore, from line 9 to line 41, although more “explicative” the information is redundant. The results may present those results the authors consider important between working conditions and voice disorders. For instance, daily class hours, daily teaching hours daily hours outdoors with student were statistically associated in the univariate analysis. However, they were not mentioned in the text.
- Discussion – prevalence of voice disorders. Although the authors compared their results with previous studies, I am missing in this section the “discussion” of the agreements and/or disagreements with previous studies. I mean why do the authors think are these results similar and/or different to previous studies?
As of now, there is no new information for the topic of work-related voice disorders among teachers in this subheading of the discussion.
- Discussion - Voice Disorder Comparison Between Kindergarten and Elementary School Teachers. Although the heading implies a discussion about voice disorders prevalence, this section is about voice disorders and working conditions, such as humidity, voice usage during teaching. Please adjust.
- Discussion – in general terms I think this section are too descriptive and there is still a lack of argumentation and discussion of the results. There are several studies on voice disorders among teachers, and the work-relatedness of voice disorders among teachers have been widely reported. Therefore, I was expecting to find in this section the argumentation on how this study fills in some lack of information on this topic. However, that was not the case.
Reviewer 2 Report
General remarks
The topic of this manuscript is of importance and highly suitable for the scope of the journal. The manuscript is very well written. The study has been planned well, the number of subjects is good, the results are presented clearly in tables and figures. Discussion is well structured and takes up relevant factors and concerns adequately also limitations of the study. The results offer possibility to give recommendations concerning educational policy. I find the manuscript suited for publication with just minor revisions.
Please find below more specific remarks.
Abstract, line 14: I suggest to exclude the word ‘However’ (I don’t see the relevance of it here.)
Line 18: Maybe it would be clearer if you used N to refer to the total number of participants and n to refer to the number of participants in the two subgoups.
Line 21: please add ‘were’ (were used)
Line 24: I suggest: …while that in…was significantly higher
Line 27: Teaching with a raised voice has already been mentioned earlier.
Line 32: were associated
Line 34: ‘teachers’ number of basic subjects’ is a bit difficult expression to understand. Maybe it would be better to say: number of teachers in (or of) basic subjects? I am not a native speaker of English, though.
Key Words: It is not necessary to repeat any expression that is already in the title.
Throughout the manuscript: Please, check that all your reference numbers are placed in brackets [ ].
Page 3, Measurements, line 27: Maybe it would be clearer to state: Based on previous studies and pre-interviews of the teachers participating this study…
Mainly all factors included in the questionnaire sound very relevant, but for some topics I would like to see a short motivation: Why you consider that ‘falsetto’ would cause vocal fatique? I would rather expect the opposite, if we mean by falsetto such voice use where the vocal folds barely collide to each other during vocal fold vibration. Maybe you mean high pitched and strained voice use by this ‘falsetto’? Please, explain shortly.
Likewise, please state shortly what is meant by raising the voice. Do you mean increasing pitch, loudness or both? (Typically these characteristics do increase in synchrony).
I wonder why you think that marietal status is relevant when considering vocally loading factors. Can you briefly explain?
Page 3, line 45: please, change word order: not to have.
Page 3, lines 46-47: I wonder what the difference is between “daily class hours” and “daily teaching hours”. Could you shortly clarify.
Table 1: please correct a typographic error: should be ‘single’ (instead of singer)
Table 2: Maybe the table would look nicer if you put ‘yes’ and ‘no’ exactly one under the other.
Instead of vocal ‘cord’ I recommend vocal fold (cord refers to the ligament).
Could you please define what is meant by vocal cord congestion? Redness? Swelling?
Page 6, lines 19-22, please add Figure 1 a, b, c.
Throughout the paper, I wonder whether ‘pupil’ would be more suitable word than ‘student’.
Page 8, line 7: I suggest a slight rephrasing: that more than half of the participants selected to have a negative impact on their voices.
Line 12: more than half
Table 4, please correct the fact that some words have been cut in a funny way.
Here also: single instead of singer.
Discussion, page 12, lines 4- 5: I recommend that you make a new sentence: However, the prevalence of voice disorders was even higher in Egypt being 80,1 %...
Line 10, please add ‘in’ (far higher than in a similar study)
Page 13, line 7: Discussion about air humidity. Please specify whether you are taling about dryness of air or maybe excessive humidity.
Throughout the manuscript: Instead of voice usage(s) I recommend: types of voice usage.
Page 13, lines 16-17, I suggest a change in word order: the risk was 2,47 times higher…
Line 17: Instead of ‘the same’ results I recommend ‘similar results’.
Page 14, line 5: “…with the exception of the VHI scale…” However, also VHI scale is subjective. Perhaps it would be clearer to say With the exception of the VHI scale which is standardized,…
Page 14, line 29: Again here please rephrase, e.g. ‘number of teachers in basic subjects should be increased’.
Reviewer 3 Report
The objectives of this study were to determine the prevalence of voice disorders and the associated factors in early childhood teachers, and to determine if differences exist between kindergarten and elementary school teachers. This research has a well-established theoretical background in the introduction. Also, research methods are appropriate. If authors make some changes, I think it would be a valuable study.
1. Line 26, Measurements: In this study, there are 11 voice disorders, which criteria should be described.
2.Duplicate title in Figure 1.
3. The authors need to increase the resolution in Figure 1.
4. Were 'voice symptoms' divided by subjective judgments from teachers? In the 'Method' section, describe the definition of variables in more detail.
5. Some fonts is different in Table 1. Please modify the font.
6. In Table 1, the significance level of 'education' is indicated as <0.001. The significance level needs to be corrected by the ***mark shown in footnote.
7. In this study, 'Married' included all married, wide and divorced, but shouldn't we distinguish them? When using Married as a 'demographic variables' in epidemiological studies, married, wide, and divorced are generally distinguished. But in this study, Married is not an important variable. Therefore, in the Methods section, authors must add a definition of a confounding variable.
8. The "***" sign in Table 2 is unlikely to be necessary.
9. Delete the lines shown under "Work organization" in Table 3.
10. In Table 3 and 4, all significant levels <0.001 should be changed to ***notation.
11. Table 4 is too complex. I think we'd better separate the table 4.
12. Page 11, Line 6-41: "Compared with the unadjusted value (OR = 1.62; 95% CI, 1.09–2.04; p = 0.018), the strength of the association between the school type and voice diseases decreased gradually and the significance disappeared after model 1 to model 4 adjustments. The adjusted model 1 ..."
- The result is too lengthy. Is it necessary to describe all regression models in this study? I think the most important models are the unadjusted model and model 4. In the Results section, it would be best to explain only two models.
13. Page 11, Line 43: It would be better to delete the sub-title in the discussion section.
